# Diversity Analysis and Genetic Relationships among Local Brazilian Goat Breeds Using SSR Markers

**DOI:** 10.3390/ani10101842

**Published:** 2020-10-10

**Authors:** Marcos Paulo Carrera Menezes, Amparo Martinez Martinez, Edgard Cavalcanti Pimenta Filho, Jose Luis Vega-Pla, Juan Vicente Delgado, Janaina Kelli Gomes Arandas, Laura Leandro da Rocha, Maria Norma Ribeiro

**Affiliations:** 1Departamento de Agropecuária, Universidade Federal da Paraiba, Bananeiras 558220-000, Brazil; marcoscarrera@cchsa.ufpb.br; 2Departamento de Genetica, Universidad de Córdoba, 14071 Cordoba, Spain; amparomartinezuco@gmail.com (A.M.M.); juanviagr218@gmail.com (J.V.D.); 3Departamento de Zootecnia, Universidade Federal da Paraíba, Areia 58397-000, Brazil; edgardpimenta@hotmail.com; 4Laboratorio de Genética Molecular, Servicio de Cría Caballar y Remonta, 14071 Córdoba, Spain; jvegpla@oc.mde.es; 5Departamento de Zootecnia, Universidade Federal Rural de Pernambuco, Recife 55171-900, Brazil; janaina_arandas@hotmail.com (J.K.G.A.); laura_rocha77@yahoo.com.br (L.L.d.R.)

**Keywords:** genetic conservation, molecular markers, genetic distance, heterozygosis, animal genetic resources

## Abstract

**Simple Summary:**

This study aimed to evaluate the genetic diversity of six groups of native Brazilian goats using a panel of single sequence repeats (SSRs). Results indicated a definite genetic differentiation among the Brazilian goat herd, which indicates the existence of at least four breeds according to the international concepts (Moxotó and Repartida; the Grauna and Serrana Azul; Canindé and Marota breeds).

**Abstract:**

The genetic diversity of six Brazilian native goats was reported using molecular markers. Hair samples of 332 animals were collected from different goat breeds (Moxotó, Canindé, Serrana Azul, Marota, Repartida, and Graúna) from five states of Northeast Brazil (Paraíba, Pernambuco, Rio Grande do Norte, Bahia, and Piauí). A panel of 27 microsatellites or single sequence repeats (SSRs) were selected and amplified using a polymerase chain reaction (PCR) technique. All populations showed an average allele number of over six. The mean observed heterozygosity for Brazilian breeds was superior to 0.50. These results demonstrated the high genetic diversity in the studied populations with values ranging from 0.53 (Serrana Azul) to 0.62 (Repartida). The expected average heterozygosity followed the same trend ranging from 0.58 (Serrana Azul) to 0.65 (Repartida), and the values obtained are very similar for all six breeds. The fixation index (Fis) had values under 10% except for the Moxotó breed (13%). The mean expected heterozygosity of all Brazilian populations was over 0.50. Results indicated a within-breed genetic variability in the Brazilian breeds based on the average number of alleles and the average observed heterozygosity. The interbreed genetic diversity values showed proper genetic differentiation among local Brazilian goat breeds.

## 1. Introduction

Many domestic species originating in Europe were introduced to America in the colonial period. Brazilian goat breeds are derived mainly from Portuguese settlers with animals since the 16th century [1,2]. The Brazilian goat breeds were developed in the national territory due to natural and artificial selection promoted by smallholders, emphasizing the morphology and fitness traits and other associated processes like genetic drift [1]. They developed unique traits such as rusticity, prolificacy, and disease resistance [3], becoming a vital economic resource to household communities.

In addition to early Iberian introductions, Asian breeds (Bhuj, Jamnapari, Mambrina, and Angorá) were introduced. Recently, goats of European breeds (Alpine, Saanen, Toggenburg, and Murciano-Granadina) and African breeds (Anglo-Nubian and Boer) have been imported into Brazil to improve milk and meat productions.

Brazil has more than ten million goat heads, according to FAOSTAT [4]. Approximately 90% of the national herd consists of animals with no defined breed pattern (NDBP) originating from indiscriminate crossbreeding between foreign and local breeds. However, there are two officially recognized local breeds (Moxotó and Canindé), and other local ecotypes (Serrana Azul, Repartida, Marota, Graúna, and Gurguéia) named according to their origin region or a particular trait. Otherwise, since 2015, these genetic resources are officially recognized as locally adapted breeds, local, or autochthonous breeds (Biodiversity Law. Law No. 13.123/15) [5]. According to a new international breed classification system, local breeds occur only in one country [4].

The local Brazilian goat breeds Moxotó, Canindé, Serrana Azul, Marota, Repartida, and Graúna, are animals of multiple functions and adapted to the climatic conditions and smallholder farming systems. They are rustic animals with less nutritional requirements, raised extensively, and survive in areas where they predominately forage foe food with low nutritional value, unsuitable for foreign breeds. These breeds are the only source of animal protein for low-income populations in the semiarid northeast. Despite the historical and social importance of these breeds, they have been undervalued for decades due to their low contribution to the national economy and their concentration in the most impoverished Brazilian region. This fact has contributed to limited scientific knowledge about these valuable genetic resources.

Another risk factor is that in the last years, the introduction of highly specialized foreign breeds in the Brazilian herds has caused a fast replacement and genetic erosion of the local ones, as observed by [6] and [7]. This situation requires conservation programs to protect the remaining genetic diversity in these local populations. Since the maintenance of these breeds is essential to guarantee an appropriate level of livestock biodiversity and the smallholder family’s maintenance.

Accurate knowledge of genetic resource diversity is fundamental for properly targeting conservation strategies and using these resources. Genetic characterization of animals through molecular markers had it shown efficient to quantify genetic variation in different populations.

This study aimed to use molecular markers (microsatellites) to determine the genetic variation between local Brazilian goats.

## 2. Materials and Methods

Random samples were collected from 332 Brazilian goats: 60 Moxotó, 50 Canindé, 55 Serrana Azul, 68 Marota, 52 Repartida, and 47 Grauna (Appendix A). DNA was extracted from 10 selected hairs from each animal according to the methodology proposed by Walsh et al. [8] using the same 27 single sequence repeats (SSRs) studied by Menezes et al. [9] (Appendix A) were amplified and PCR products were separated by electrophoresis in an automatic sequencer ABI 377XL (Applied Biosystems, Foster City, CA, USA).

The PCR products were accomplished both by the internal size standard and by the same reference sample to correct the few variations in allele size assignation among runs. GENETIX v4.04 software [10] was used to calculate allele number, observed, and unbiased expected heterozygosity estimates within breeds [11]. Distribution of gene variation within and between breeds was estimated, according to Wright [12] F-statistics using Weir and Cockerham’s [13] method. The genetic distance, DA [14], was estimated using the POPULATIONS 1.2.28 [15] computer program. Distances between each pair of populations were used to build a UPGMA tree [16], and a Bootstrap resampling test (n = 1000) was performed to verify the dendrogram robustness. Factorial Correspondence Analysis [17] was performed to test the possible admixtures between the populations using the module “AFC sur populations” of the GENETIX v4.04 software.

## 3. Results and Discussion

Genetic variation within breed and fixation index (F_is_) of Brazilian goat breeds are presented in Table 1. All populations showed an average allele number of over six. The mean observed heterozygosity for Brazilian breeds was higher to 0.50. These results demonstrated the high genetic diversity in the studied populations with values ranging from 0.53 (Serrana Azul) to 0.62 (Repartida). The expected average heterozygosity followed the same trend ranging from 0.58 (Serrana Azul) to 0.65 (Repartida), and the values obtained are very similar for all six breeds. The fixation index (F_is_) had values under 10% except for the Moxotó breed (13%).

The mean expected heterozygosity (H_e_) of all Brazilian populations was over 0.50. The Serrana Azul was the breed with the lowest value (0.53). The matrix of D_A_ genetic distances [14] among Brazilian breeds and F_ST_ among population pairs are showed in Table 2. The smallest genetic distance observed was between Serrana Azul and Graúna breeds, and the highest genetic distance between Serrana Azul and Canindé breeds. This situation was similar to the F_ST_ values.

The average DA among all populations was 0.122, which indicates a clear split of the different Brazilian breeds as confirmed by the average FST value (0.075). The differences can be observed in the dendrogram built through the UPGMA method (Figure 1). Two different groups were identified: the first formed by Repartida, Moxotó, Canindé, and Marota. Serrana Azul and Graúna formed the second group. The Bootstrap values ranged from 0.59 to 0.95, showing the degree of confidence in the topology.

The numbers on the nodes are percentage bootstrap values from 1000 replications of resampled loci. The factorial analysis (Figure 2) showed more than 54% of the total variation in the two first axes, indicating high confidence and genetic relationship among populations. A study conducted using 13 microsatellites with six indigenous Iranian goat populations observed two main groups based on the phylogenetic tree and FCA analysis [18].

Ibnelbachyr et al. [19], evaluating the genetic differentiation of Moroccan goat breeds using 12 microsatellite DNA markers, found 64% of the total variation in the first three factors, whereas Bulut et al. [20] evaluating the genetic diversity of eight domestic goat populations raised in Turkey with 11 microsatellites loci, obtained 47% of the total variation of the first three factors.

The average number of alleles was higher than those found in different studies with goats in several regions globally, including some breeds in the Americas [21] and the Chinese breed of Daiyun goat [22]. However, the highest number of alleles was reported above 7 in other goat populations in Turkey [20], Marrocos [19], and Saudi Arabia [23]. The higher number of alleles presented by these studies, according to the authors, is justified by the gene flow, the location of the countries on migration routes, and the proximity of the initial centers of domestication, being the vast allelic diversity found very useful for selection. Differences in the mean number of alleles can also be influenced by the sample size and the number and type of microsatellites used.

Recent studies evaluating the genetic diversity of Brazilian goat breeds using microsatellite markers observed high levels of observed and expected heterozygosity and suggested substantial genetic diversity in the evaluated breeds.

The average heterozygosity observed in our study was higher than that reported by Silvestre et al. [24] (0.385) in Nubian goats in Brazil. On the other hand, Hossam Mahmoud et al. [23] obtained mean heterozygosity higher than 0.91, in the local Saudi Arabia goat breed.

According to Jianmin and Wenbin [25], in improved breeds, introgression and gene exchange are widespread, mainly increasing heterogeneity in the population.

Only in the Moxotó breed, the F_IS_ value was higher than 0.10. The F_IS_ values in the studied populations shown genetic homogeneity. The high F_IS_ value in the Moxotó breed (0.13) may be due to structure with subdivision into herds and low gene flow among the herds. Our samples were collected from four different herds that were geographically and reproductively isolated, and each of these populations is generally closed with a reduced number of sires, which favors inbreeding. In conservation programs, inbreeding levels should be taken into account, mainly when the same male is used for a prolonged period.

The low F_IS_ values obtained in our study were contrasting with those found by other authors in goat populations [24,26,27,28].

The degree of genetic differentiation (F_ST_ = 0.075) among all populations was predictable, recent studies have already indicated this degree of differentiation of our Brazilian goats [1], and Frankham et al. [29] suggest that F_ST_ values between 0.05 and 0.3 are typical of differentiated livestock breeds. In the European goat breeds, the F_ST_ lies around 5%.

The genetic makeup of the Brazilian goat breeds has been influenced by breeds of the Iberian Peninsula and even West African [21,30,31,32], promoting gene flow between the Iberian Peninsula, northern Africa, and the Canary Islands and Cape Verde [33,34,35]. Currently, Brazilian goat breeds are genetically distinct with a clear differentiation even from their possible ancestors [1,6,36,37,38]

As pointed out before, the Brazilian goat herds are very concentrated in one specific region, so that gene flow among breeds must be shared, especially considering that the herd book is either not ongoing or is very rudimentary. Despite this, close breeding management produces some “family” effect within breeds. In general, the average number of alleles and the mean heterozygosity obtained in this work suggest a high genetic diversity within the Brazilian goat breeds. These breeds are also targeted to crossbreeding with exotic and with each other Brazilian local goat breed, as reported by Rocha et al. [6].

The use of crosses within the local breeds is widespread in Brazil, as observed in our research (Figure 1). Those with the exotic breeds were sufficient to consider establishing a genetic conservation program for Brazilian indigenous goat populations.

Therefore, efficient herd management is vital to the conservation of these genetic resources by avoiding breeding with related individuals, exchanging individuals among herds, and increasing the effective number.

The UPGMA dendrogram (Figure 1) constructed with the D_A_ genetic distance values grouped Serrana Azul and Graúna and performed another group joining Moxotó, Repartida, and Marota. The proximity of Moxotó and Repartida could be explained by the genetic composition of the Repartida ecotype, which arose by segregation within Moxotó, differing only in coat color.

Despite the geographical distance among the Moxotó and Canindé, these breeds probably shared the same gene pool.

The correspondence factorial analysis showed the same distribution of the populations, but in this case, the Marota and Canindé breeds were distant from the others. It was not possible to separate Moxotó and Repartida breed, and this fact was well explained before.

The Marota have had a higher genetic distance with each other breeds and, it could be due to a genetic drift effect promoted by founder effect and geographic isolation. There is a high similarity between the Serrana Azul and Grauna breeds. There was gene flow between the two groups favoring a smaller genetic distance among them. The breeders have only recently separated these populations based on the coat color pattern, and there has not been a genetic differentiation other than the few loci involved in the coat color determination.

## 4. Conclusions

A significant genetic differentiation among the Brazilian goat breeds was verified, which indicates the existence of at least four breeds according to the international concepts (Moxotó and Repartida; the Grauna and Serrana Azul; Canindé and Marota breed).

The genetic diversity and differentiation among the local Brazilian goat breeds must be safeguarded through in-situ and ex-situ conservation actions, integrated into local development programs.

## Figures and Tables

**Figure 1 animals-10-01842-f001:**
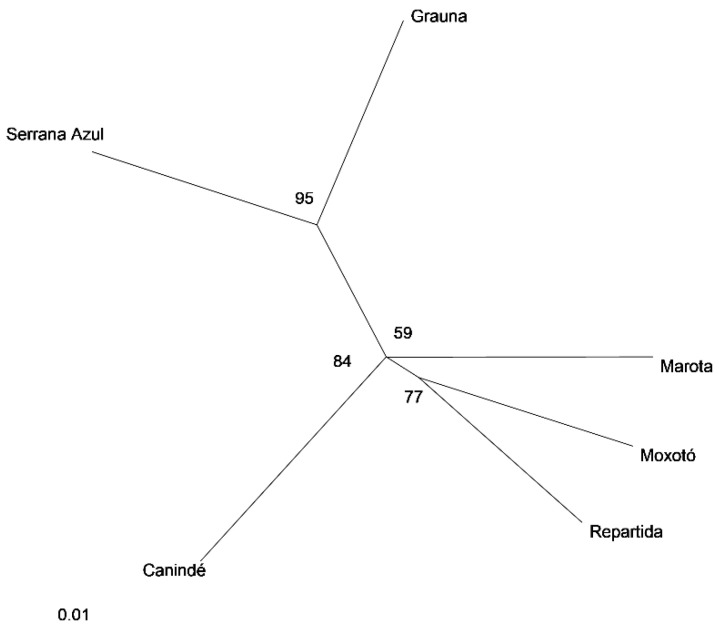
UPGMA dendrogram of Brazilian breeds, based on D_A_ distance [14].

**Figure 2 animals-10-01842-f002:**
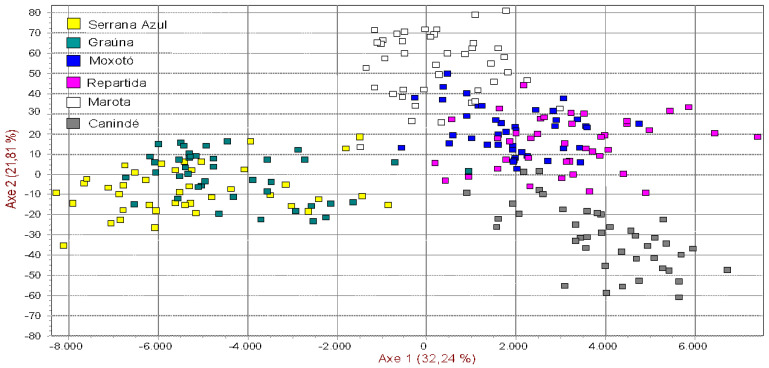
Spatial representation of Brazilian breeds using factorial correspondence analysis.

**Table 1 animals-10-01842-t001:** The number of samples (n), the average number of alleles (Na), average observed heterozygosity (Ho), the average expected heterozygosity (He), and F_IS_ values of the Brazilian breeds.

Populations	n	Na	Ho	He	F_IS_ (IC = 95%)
Serrana Azul	55	6.22	0.53	0.58	0.06 (−0.01–0.09)
Moxotó	60	6.29	0.55	0.62	0.11 (0.06–0.15)
Marota	68	6.07	0.58	0.61	0.04 (−0.02–0.07)
Canindé	50	6.15	0.60	0.64	0.05 (−0.01–0.08)
Repartida	52	6.70	0.62	0.65	0.07 (0.02–0.09)
Graúna	47	6.56	0.61	0.62	0.03 (−0.03–0.07)
Overall	332	6.32	0.58	0.62	

**Table 2 animals-10-01842-t002:** Matrix of genetic distance DA between pairs of Brazilian breeds (below diagonal) and F_ST_ between pairs of populations (above diagonal).

Populations	Serrana Azul	Moxotó	Marota	Canindé	Repartida	Graúna
Serrana Azul		0.084	0.087	0.105	0.104	0.036
Moxotó	0.124		0.072	0.045	0.051	0.073
Marota	0.143	0.110		0.097	0.082	0.066
Canindé	0.145	0.100	0.144		0.047	0.089
Repartida	0.140	0.086	0.121	0.097		0.085
Graúna	0.082	0.130	0.130	0.144	0.139

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
