# Peer review of "Diversity Analysis and Genetic Relationships among Local Brazilian Goat Breeds Using SSR Markers"

_animals, 2020, doi:10.3390/ani10101842_

Round 1

Reviewer 1 Report

Dear authors

I appreciate that you have followed my suggestions, however I believe they have not been completely satisfied. Furthermore, new corrections are required following the changes made. In particular:

  • it would be useful to indicate the numerical consistencies of the studied breeds
  • It would be interesting to state the advantages they provide in breeding when compared to other goat breeds reared in Brazil
  • The authors wrote that “- It was verified a significant genetic differentiation among the Brazilian goat breeds, which indicates the existence of at least four breeds according to the international concepts”; please describe this “international concept” in Introduction or in Results and Discussion .

Line 48: Change the reference with the appropriate number

Line 48: Please change “They acquire…” with “They developed…”

Line 49: Please change “becoming vital….” With “ becoming an important economic resources…”

Line 51: Please change “…goats from Asian breeds…” with “…goats of Asian breeds…”

Line 53-54: Please change “…In recent decades have seen introductions of breeds imported from Europe (Alpine, Saanen, Toggenburg, and Murciano-Granadina) to expand the milk production and African breeds (Anglo-Nubian and Boer) were imported to improve the meat production in Brazil.” with “Recently, goats of European breeds (Alpine, Saanen, Toggenburg, and Murciano-Granadina) and of African breeds (Anglo-Nubian and Boer) have been imported in Brazil in order to improve milk and meat productions”

Line 64: there is an extra space

Line 65: a space is missing

In general there are extraspaces and missing spaces throughout the menuscript

Line 74: Please change “….is fundamental to for proper targeting…” with …is fundamental for proper targeting…”

Line 77: Please change “…to determine a genetic…” with “…to determine the genetic…”

Line 149-150: Please check this sentence, the meaning is not clear to me

Line 145-148: Please check this sentence, the meaning is not clear to me.

Line 153: Please change “…Our sample was…” with “… Our samples were..”

Line 161-162: Please check this sentence, the meaning is not clear to me.

Line 168-170: Please check this sentence, it doesn’t sound good

The final sentence of the Result and Discussion Section is repeated in Conclusion, please change it in one of the two sections.

Reviewer 2 Report

I have no more comments.

Author Response

The second reviewer didn´t suggest more review.

This manuscript is a resubmission of an earlier submission. The following is a list of the peer review reports and author responses from that submission.

Round 1

Reviewer 1 Report

Dear authors

Local breeds are of great interest for scientific community, their safeguard is essential to ensure a fair level of livestock biodiversity.

To this purpose the knowledge of their genetic structure and of their genetic peculiarities is necessary.  For this reason I consider studies like yours if great interest for scientific community.

However, I believe that the work needs some changes in order to be published in a journal like Animals.

Simple summary:  must be improved stating clearly the aim of the study and the main result.

Introduction: the first sentence “Many genetic populations of domestic animals were introduced in America, coming from Europe” sounds very bad, please modify it. In this section a short description of the goat breeds analysed in this study would be very appeciable, a figure (eventually as supplementary material) showing a subject of each breed would be appreaciable too.

Materials and methods: Please eliminate the list of microsatellites used in the main text and provide a table as supplementary material in which it is indicated the name of the microsatellite, the primers used, the repetition, the minimum and maximum sizes and chromosome in which it is located.

Results and discussion:

Line 155: Please motivate the sentence “The degree of genetic differentiation (FST=0.075) among all populations was predictable,”

Conclusions: this section needs to be improved

In general check the references in the text: for example line 144: “According to Wang and Yue [26] in improved breeds, introgression…” the references [26] correspond  to “Jianmin, W.; Wenbin, Y. Genetic relationships of domestic sheep and goats in the lower reaches of the Yellow River based on microsatellite analysis. Biodivers. Sci. 2008, doi:10.3724/sp.j.1003.2008.07050.”.

Author Response

Please see the a attachment

Reviewer 2 Report

General comments

First, it is not clear the main purpose of this study; why conserving those breeds are important in Brazil. They were not native animals in South America and were brought mainly from Europe. And then they are disappearing because they are not so economically important, being replaced by newly introduced breeds. What is the problem? Authors need to justify why old breeds need to be conserved. If they are better adapted in each local environment, this is an interesting topic to study. Showing how they are adapting better than new breeds is very important.

Second, the analysis is straightforward. Authors need to explain why showing genetic diversity or similarity among breeds is so important when conserving those breeds.

Details

Line 46: “400” => “four”.

Lines 170-172: Are there any indigenous (native) goat populations other than ones brought from Europe 400 years ago? Is the proposed conservation program to protect native goats (if they exist) or those breeds brought 400 years ago?
